# Which Is the Best Treatment in Recurrent Thymoma? A Systematic Review and Meta-Analysis

**DOI:** 10.3390/cancers13071559

**Published:** 2021-03-29

**Authors:** Marco Chiappetta, Ugo Grossi, Isabella Sperduti, Stefano Margaritora, Giuseppe Marulli, Alfonso Fiorelli, Alberto Sandri, Tetsuya Mizuno, Giacomo Cusumano, Masatsugu Hamaji, Alfredo Cesario, Filippo Lococo

**Affiliations:** 1Università Cattolica del Sacro Cuore, 00168 Rome, Italy; marco.chiappetta@policlinicogemelli.it (M.C.); stefano.margaritora@unicatt.it (S.M.); 2Thoracic Surgery, Fondazione Policlinico Universitario A. Gemelli IRCCS, 00168 Rome, Italy; 3Regional Hospital Treviso, DISCOG, University of Padua, 31100 Treviso, Italy; grossiugo@gmail.com; 4Biostatistics, Regina Elena National Cancer Institute, IRCCS, 00100 Rome, Italy; isabella.sperduti@ifo.gov.it; 5Thoracic Surgery Unit, University Hospital of Bari, 70124 Bari, Italy; beppemarulli@libero.it; 6University of Campania Luigi Vanvitelli, I-80138 Naples, Italy; fiorelli.alfonso@gmail.com; 7Department of Thoracic Surgery, San Luigi Hospital, Regione Gonzole 10, Orbassano, 10043 Torino, Italy; sandri.alberto@gmail.com; 8Division of Thoracic Surgery, Shizuoka Cancer Center, Shizuoka 411-8777, Japan; t-mizuno@aichi-cc.jp; 9Policlinico Vittorio Emanuele Hospital, 95123 Catania, Italy; giacomare55@hotmail.com; 10Department of Thoracic Surgery, Kyoto University Hospital, Kyoto 602-8566, Japan; mhamaji@kuhp.kyoto-u.ac.jp; 11Open Innovation Manager, Direzione Scientifica, Fondazione Policlinico Universitario A. Gemelli IRCCS, 00168 Rome, Italy; alfredo.cesario@unicatt.it

**Keywords:** meta-analysis, recurrent thymoma, surgery, chemotherapy, radiation therapy

## Abstract

**Simple Summary:**

Thymomas are rare tumors developing in the anterior mediastinum. Despite their usually indolent behavior, recurrence might occur in 5–15% of cases. Considering their rarity, the optimal recurrence treatment is still unclear even if surgical treatment seems to ensure a remarkable long-term survival compared to chemo- or radiotherapy. However, the major part of studies report low patient numbers, and it is difficult to plan prospective studies due to tumor characteristics, long follow-up and rarity of cases. For these reasons, we planned a systematic review and meta-analysis comparing surgical treatment with other therapies, in order to identify the best treatment for these patients. Our meta-analysis included more than 700 patients showing that surgical treatment seems to be associated with a better survival compared to other treatments and should be considered where feasible.

**Abstract:**

Background: Optimal recurrent thymoma management remains arguable because of limited patient numbers, and relatively late and variable recurrence patterns. Given the absence of high-quality evidence and relatively small study cohorts, we performed a quantitative meta-analysis to determine the outcome of surgical and non-surgical approaches assessing the five-year overall survival (5y overall survival (OS)) in patients with recurrent thymoma. Methods: We performed a comprehensive literature search and analysis according to PRISMA guidelines of studies published from 1 January 1980 until 18 June 2020 from PubMed/MEDLINE, EMBASE, and Scopus. We included studies with the cohorts’ superior to 30 patients describing recurrent thymoma treatment, comparing surgical and non-surgical approaches reporting survival data. Results: Literature search revealed 3017 articles. Nine studies met all inclusion criteria and were selected for the meta-analysis. The recurrences were local/regional in 73–98% of cases and multiple in 49–72%. After treatment, 5y OS ranged from 48–77% and 10y OS from 37–51%. The quantitative meta-analysis showed a better outcome comparing surgical vs other treatments. Two studies showed statistically significant risk differences in the 5y OS favoring complete resection. After pooling results of seven studies using the random model, the combined 5y OS risk difference was 0.39, with lower and upper limits of 0.16 and 0.62, respectively (*p* = 0.001), and a moderate heterogeneity among studies (*p* = 0.098, I2 = 43.9%). Definitive conclusions could not be drawn regarding the prognostic impact of myasthenia gravis, histology, and patterns of relapse reported in literature. Conclusions: Surgical treatment after thymoma recurrence is associated with a significant better 5y OS; therefore, surgical resection should be preferred in all technically feasible cases.

## 1. Introduction

Thymomas are relatively rare tumors of epithelial thymic cells representing approximately 0.2–1.5% of all malignancies [1]. Surgical resection remains the mainstay of treatment, facilitating long-term survival [2]. Thymomas have usually an indolent behavior; however, the natural history is often unpredictable. Indeed, recurrences are found in 10–30% of patients after radical resection (R0) and might occur even after 10 to 20 years [2,3,4]. As largely documented [3,4,5], thymoma recurrences are more often loco-regional [6] rather than hematogenous (distant) and usually involve the pleura or mediastinum. In the majority of cases, distant metastatic spread affects the pulmonary parenchyma; other organs are usually spared [2].

Clear evidenc regarding the best therapeutic option for recurrent thymoma are scarce because of the rarity and relatively late onset of recurrence. Moreover, few studies have focused on the management of recurrent thymoma, and no randomized clinical trials have been conducted yet. Accordingly, no evidence-based guidelines have been formulated so far.

Several studies reported repeated resections in recurrent thymoma patients, which showed improved early- and long-term outcomes [2,3,4,5], whereas other studies are not in favor of re-do surgery and support chemotherapy and/or radiotherapy with different protocols and schemes [7,8].

Based on this knowledge gap, we performed a systematic review and meta-analysis focusing on survival data.

## 2. Materials and Methods

The authors developed the study protocol detailing pre-specified methods of the analysis and eligibility for the review in accordance with 2009 PRISMA guidance [9].

### 2.1. Eligibility Criteria

#### Study Characteristics

Study details were defined using the PICOS framework (Population Intervention Comparison Outputs Study). Search term definitions were largely inclusive, promoting a sensitive search of studies reporting interventions for recurrent thymoma (Appendix B).

Population: The review aimed to identify studies, which included patients, who underwent surgical interventions with the primary intent to treat recurrent thymoma. Cohorts of minimum 30 patients were eligible. This threshold was taken to exclude case reports and small case series, which often reported a single surgeon’s personal experience or early experiences of experimental procedures. Only studies reporting data regarding the first recurrence outcome were included. Studies that reported survival analysis, including thymic carcinoma, were only selected if separate survival analysis for thymomas was present.

Intervention: Surgical and non-surgical therapeutic approaches.

Comparisons: Studies were eligible regardless of whether they were retrospective or prospective in design, controlled or uncontrolled.

Outcomes: Studies were eligible if they provided data on survival comparing surgical vs non-surgical strategies. Non-surgical treatments were defined as chemo- or radiotherapy alone or in combination.

### 2.2. Report Characteristics

Year of publication: Any publication date starting from 1 January 1980 was eligible until 18 June 2020.

Language: Only studies with full text in English language were included.

Type of study: Only peer-reviewed publications reporting primary data were eligible. Therefore, reviews, editorials, letters and other forms of secondary expert opinion were excluded at the screening stage. Only full manuscripts were eligible excluding conference abstracts and proceedings. No constraints were imposed, based on the level of evidence.

#### Information Sources, Study Selection and Data Collection

Three authors (MC, UG, and FL) performed a comprehensive search of the literature on PubMed, Scopus, and Evidence Based Medicine reviews (including the Cochrane database of systematic reviews and the Cochrane central register of controlled trials).

A lack of eligibility, resulting in exclusion for analysis, was determined by two authors (MC and FL) who read the abstract of the study. Full-texts of all remaining studies were obtained and assessed by the same reviewers blinded to the studies’ title, authors, institutions, and other publications. A third senior author (AC) decided in case of eventual disagreements regarding the inclusion of a particular study. The authors were contacted in instances of doubt regarding result repetition, and this was performed on 2 occasions.

Two authors (MC and FL) extracted the following data onto a Microsoft Excel spreadsheet: Study characteristics and outcomes (type of treatment(s), recurrence characteristics, disease free interval, completeness of resection, administration of adjuvant treatments, primary thymoma characteristics, presence of myasthenia gravis, overall survival, follow-up). The final manuscript was shared with the different principal investigators of eligible studies (co-authors of the present study) and the final manuscript was approved by all co-authors. Quality assessment of studies retained for full-text review were independently evaluated by the same authors (MC and FL) using the Joanna Brigs Institute (JBI) critical appraisal tools for analytic observational studies [10]. The JBI critical appraisal tool has 11 items to assess cohort studies.

### 2.3. Statistical Analysis

Rate differences and standard error were quantitatively synthesized using the Comprehensive Meta-Analysis Software, version v.2.0 (CMA, Biostat, Englewood, NJ, USA).

Five-year OS was calculated from the date of recurrence treatment until the date of death by any cause in all seven studies. For each study, the standard error of the 5-year or 10-year mortality rate was calculated according to the formula,
SE = √(a × (1 − a)/√(*n*)
where a = 5-year rate and *n* = sample size.

The difference in 5-year (5yr) survival between the surgical and the non-surgical group was calculated for each study. The standard error (SE) of the difference was calculated as SE (diff) = √((SE1)2 + (SE2)2) where SE1 is the SE of the 5yr survival rate of the surgically managed patients and SE2 is the SE of the 5yr survival rate of non-surgically managed patients.

Statistical significance was set at *p* < 0.05. The I2 and Q statistics were used to test statistical heterogeneity among the studies included. An I2 value above 50% was considered representative of considerable heterogeneity across the included studies. Independently of the heterogeneity degree observed, a random-effect model was identified as the most appropriate approach. The results were also derived from a fixed-effect model. A funnel plot was used for publication bias assessment.

## 3. Results

### 3.1. Literature Results

In the comprehensive literature search on PubMed/MEDLINE, Scopus and EMBASE, 3017 articles were found. By reviewing titles and abstracts, 2354 articles, formatted as reviews, editorials, letters, commentaries or case reports, and additional 621 duplicates or non-English language articles were excluded. Forty-two eligible studies were selected and retrieved in full-text version; no additional study was found by cross-reference.

Thirty-three full text reports were excluded for the following reasons: lack of direct overall survival comparison between surgery and other therapies (N = 22 studies), sample dimensions inferior to 30 patients (N = 11 studies).

Finally, 9 studies met all inclusion criteria and were selected for meta-analysis [2,4,5,11,12,13,14,15,16] (Figure 1). Of note, two studies were from our institution [2,15]. The quality assessment of the studies is reported in Appendix A.

The characteristics of the eligible studies are presented in Table 1, Table 2 and Table 3.

In detail, all of them were retrospective case series (multicentric in 5 cases) from Europe (6 studies) and Asia (3 studies). No randomized trials comparing surgical and nonsurgical management of recurrent thymoma were found. Seven hundred seventy-eight patients with recurrent thymoma were included in our meta-analysis.

### 3.2. Recurrence Characteristics

The site of recurrence was reported in all papers, with some heterogeneity in classification. The recurrence was local/regional [16] in the majority of cases, while distant recurrence occurred in 2–27% of cases. In detail, the first site of local recurrence was the thymus bed (13.5–27.9%) [2,4,12,14,15,16], while the pleura was the most commonly involved among regional intrathoracic sites (49.6–58.1%) [2,4,15]. Lung metastases were a common type of distant recurrences while the pleura (regional relapse) resulted as the most common recurrence site. Three studies [2,14,15] reported the number of localizations, with multiple lesions occurring in the most patients (range 49.5–71.6%). Distant recurrences were reported in 6.6–23.4% of cases [2,4,12,14,15,16] with the lung as the most common distant site involved, also if liver or bone metastases were reported in some cases [2,11]. The disease free interval from the thymectomy to the recurrence appearance ranged between 50 and 98 months.

### 3.3. Overall Survival

In all studies, the overall survival was calculated starting from the date of surgery to the data of death due to any cause. All studies reported 5yr and 10yr OS after treatment, ranging from 48% to 76.6% for 5yrOS and from 37% to 51% for 10yrOS (Table 2). All studies reported a better survival rate after surgical treatment when compared with other treatments. However, in two studies, the survival comparison was performed between patients receiving complete resection and those undergoing incomplete resection or other treatments [5,16].

### 3.4. Surgery and Other Treatments

Data regarding surgery (complete vs incomplete resection) and other treatments (chemotherapy, radiotherapy or both) for recurrences were identifiable in the vast majority of the studies (Table 2). In detail, a complete resection was achieved in 60–90% of cases, while only few patients did not receive any kind of treatment, which is likely owing to poor clinical condition or the indolent nature of the recurrence (3 patients in total) [2,12]. A comparative survival analysis was present in all studies, but only seven of them demonstrated differences between surgery and other treatments.

Margaritora et al. [2] and Hamaji et al. [11] demonstrated that surgical management is an independent prognostic factor when compared to non-surgical treatment, while Marulli et al. [14] reported a better survival in patients who underwent complete rather than incomplete resections or other treatments.

Mizuno et al. [13] reported survival advantage for surgical (also considering R1-R2 patients) compared to non-surgically treated patients. Although the difference in survival between R0-R1 was significant, the difference between R2 and non-surgically treated patients was not significant (*p* = 0.143).

Furthermore, Chiappetta et al. [15] reported survival benefits for surgical treatment compared to other treatments even if not statistically significant (*p* = 0.064), considering only the first recurrences. Instead, considering all recurrences, a survival difference was observed comparing survival after surgery with survival after chemo- or radiotherapy, with a trend in favor of complete vs incomplete resections (*p* = 0.086). In this setting, Sandri et al. [4] and Fiorelli et al. [12] identified the completeness of resection as a favorable prognostic factor compared to incomplete resection or other treatments. Finally, Bae et al. [16] and Ruffini et al. [5] reported a better survival rate in patients who underwent complete vs incomplete resection or other treatments.

#### Meta-Analysis Results

Seven studies were included in the quantitative analysis for a total of 706 patients (Figure 1 and Figure 2): of these, 476 (67.4%) received surgery and 230 (32.6%) other treatments such as chemotherapy, radiotherapy or both (see Table 1). The included studied presented clear data regarding treatment for thymomas (excluding thymic carcinomas) comparing surgery vs other treatments. Two of the individual studies showed statistically significant risk differences in 5yr OS, favoring surgery over other treatments for recurrent thymoma. By pooling the results of the seven studies, the combined 5yr OS risk difference, using the random model, was 0.39 with lower and upper limits of 0.16 and 0.62, respectively (*p* = 0.001), and with moderate heterogeneity among studies (*p* = 0.098, I2 = 43.9%). (Figure 2 and Appendix A).

### 3.5. Other Factors Affecting Survival

#### 3.5.1. Myasthenia Gravis (MG)

Data regarding myasthenia gravis were presented in all studies, with a higher prevalence in European (56–93%) compared to Asian cohorts (13.3–53.6%). Seven studies investigated the prognostic role of MG: Chiappetta et al. [15] reported MG as an independent favorable factor for OS (*p* = 0.046). Similarly, Fiorelli et al. [12] reported a better survival in MG patients at univariable analysis (*p* = 0.02); no survival differences was observed in the other five studies [2,11,16].

#### 3.5.2. Histology

All papers, except one [5], reported survival data stratified by the original WHO histology of the primary thymoma; the percentage of change in histology comparing the primary thymoma with the recurrence was also reported in five papers [2,4,12,15,16]. Such event occurred in 12–15% of cases in all studies except the paper by Sandri et al. (40% of histological upgrading) [4]. In all studies, the change in histology corresponded to a transition to a higher grade (ex. B2 of the primary thymoma to B3 in the recurrence) [2,4,12,15,16].

Bae et al. [16] reported a better, but not statistically significant survival rate in B1-B2 vs. B3 thymomas. A significant better survival at univariate, but not at multivariate analysis was reported in two studies, comparing B1-B2 vs. B3 (*p* = 0.040 and *p* < 0.001, respectively) [12,14]. Conversely, Sandri et al. [4] reported a significant better survival at univariate but not multivariate analysis, comparing A-B1 vs. B2-B3, which favored the former group (*p* = 0.03) [4]. The other studies did not report any survival difference related to histology [2,11,13,15].

#### 3.5.3. Pattern of Recurrence

All studies reported recurrence patterns with details on recurrence sites, while the analysis on the number of localizations was present in only few studies. More precisely, a better survival was reported in patients with local compared to distant recurrences in four studies only [5,12,14,16] while differences in the site of recurrence were not found in three studies [2,4,15]. Hamaji et al. [11] and Mizuno et al. [13] reported a worse survival in distant recurrences, which included thymic carcinomas. As expected, a favorable outcome considering single compared to multiple localizations was reported in three studies [2,14,15]. As reported in Section 3.2, the major part of distant recurrence were localized in the lung, which were surgically resected in more than 90% of cases [4,15].

A significant difference regarding the treatment indication was found in the paper of Mizuno et al. [13] with a significant difference in the surgical group vs. non-surgical group considering bone and liver recurrences, while no differences were present considering lung, brain or other distant localizations. However, the authors also included thymic carcinoma and did not report a comparison including thymomas only. Also Marulli et al. [14] reported that in some patients with massive vascular infiltration, surgery was excluded and, therefore, patients underwent definitive radio-chemotherapy.

## 4. Discussion

This meta-analysis promotes the role of surgical treatment for recurrent thymomas, which was confirmed by meta-analysis of data comparing surgery versus other treatments. More precisely, we found a 5 yr survival rate of 70.2–92.0% in surgically resected patients compared to 0–67.3% in those receiving other treatments; the 10 yr survival rates ranged between 29.9% and 72.5% versus 0%, and 47.9%, respectively.

The surgical treatment in recurrent thymoma patients seems to ensure a good survival outcome, even if some clinical presentations need to be considered with caution. Indeed, due to limited case availability, the prognostic impact of surgery is still questionable in case of distant or pleural diffuse localizations. Despite the major part of studies [2,4,5,11,12,14,15,16] analyzed the prognostic significance of distant metastases (but additionally included lung parenchymal localizations according to the ITMIG classification), no differences in survival were detected considering distant recurrences. Similarly, the site of relapse was associated with a worse prognosis in few studies, but without providing specifications if significant differences in terms of treatment between patients with loco-regional or distant disease were noted [5,12,14]. In the other series [2,3,11,15,16], surgically treated distant localizations did not present statistically significant differences compared to loco-regional disease presentation in overall survival.

The pleura resulted as the most common recurrence site, and surgery varies from resection of single pleural localizations to extended pleurectomies associated with hyperthermic intrathoracic chemotherapy (HITOC), as reported in other studies not included in the present meta-analysis [17,18]. The benefit of surgical treatment might be related to other aspects. Indeed, one of the most important prognostic factors reported was resection completeness, but also the administration of integrated and multiple therapies could potentially play a role [19]. More precisely, the survival advantage in case of surgical treatments seems to be associated with complete resection; a goal reached in 60–90% of the reported cases [2,4,5,11,12,13,14,15,16].

The role of debulking surgery, which in general, was not extensively addressed, remains unclear. Most studies labelled surgery with a radical intent. However, micro-macroscopic disease residuals are to some extent reported, albeit not systematically. A significantly better survival rate in complete vs incomplete resections was reported in the majority of studies [5,12,14,16], as opposed to the experiences of Sandri et al. and Chiappetta et al. The proportion of R1 and R2 resections was not specified in the two studies [4,15]. It is possible that R1 patients presented a similar survival rate than R0 patients, while R2 resections might have had an intermediate outcome between R0-R1 related to other survival treatments. This hypothesis was also described by Mizuno et al. [13] reporting a significantly better survival in surgical resections (also debulking) versus other treatments; R0-R1 thymoma patients showed a better survival rate than R2 patients, which in turn, showed a better survival rate than those treated differently, even if not statistically significant. Analyzing the differences among the reported studies, a clear discrimination between debulking vs. R0/R1 resections might explain the statistical significance when comparing the differences in these two groups of patients [2,13]. Therefore, an unspecified percentage of R1 patients might affect the significance reported in other studies, which could explain the non-significant difference in survival when compared to complete resection.

Moreover, Chiappetta et al. [15] reported that 57.7% of patients received adjuvant therapy after recurrence resection; Mizuno et al. reported a high percentage of integrated treatments before or after recurrence surgery (about 45%). Therefore, R1 patients might benefit from multidisciplinary management, which explains why similar outcomes might be expected in R0 when compared with R1 patients, and in turn, better outcomes in R2 patients.

The strategy to treat recurrent thymoma remains an intriguing issue with recognized confounding factors like the relatively indolent nature of the disease, indication and type of surgery and the use of other non-surgical options. Taking all previous considerations into account, the clinical behavior of these tumors suggests a multidisciplinary approach, where surgical resection should be performed in situations considered advanced, such as in distant recurrences and/or pleural involvement, if clinically and technically feasible. In this scenario, the role of adjuvant therapy after or prior to surgical resection in recurrent thymoma patients has not been well-investigated and was not evaluated in the present study. We can only assume that different treatments might be administered, such as neo- or adjuvant therapy for primary thymoma, pre- or post-operative chemo- or radiotherapy after recurrence resection [4,12,14,15]. For this reason, the role of integrated treatments remains difficult to assess.

The relationship between oncological outcome and MG in recurrent thymoma patients remains a highly debated issue. This issue was investigated in almost all studies included in this meta-analysis but significant differences in terms of survival emerged only in the papers by Fiorelli et al. (univariable analysis) [12] and Chiappetta et al. [15] who identified the presence of MG as a favorable independent prognostic factor, but no biological findings underlying this association are present.

Conversely, we should take into account that the presence of a combined follow-up (oncological and neurological surveillances) might lead to an early identification and treatment of recurrences. Indeed, the worsening or the re-appearance of MG symptoms might be related to anticipated exams and prompt intervention resulting in a better outcome for MG patients. However, no data in the literature support this theory and the role of MG in thymoma recurrences remains unclear.

Furthermore, the prognostic impact of the histology in these patients remains controversial; B3 histology seems to be related to a worse prognosis in three studies [2,12,16], but this was not significant at multivariate analysis. Similarly, any change in histology did not influence the prognosis in patients with recurrence [2,12,15,16]. Although it appears that the histological change shows a vector oriented towards a more aggressive histology, as diffusely reported [2,4,12,15,16].

In relation to this argument, Ciccone and Rendina [20] postulated that only the cortical part of the thymoma might be responsible for recurrence, while Bae et al. [16] theorized that the histological change might be due to the depletion of lymphocytes with subsequent increase of the epithelial component quota. The authors proposed that it could be an effect of the therapy with corticoid-steroids, noting that histological upgrading occurred only in MG patients. On the other hand, an association between histology and recurrence rate is well known [21,22], and it is possible that an epithelial and less differentiated component might be responsible for the recurrence.

### Limitations of the Meta-Analysis

The meta-analysis showed a very acceptable study heterogeneity demonstrating the robustness of these results.

However, different aspects of this meta-analysis should be considered for their translation into daily clinical practice. Two of the most controversial points on recurrent thymoma treatments concern the low quantitative dimension of the population and treatment indications. In particular, a not negligible selection bias should be taken into account in all studies: First, the risk that surgery might be indicated for resectable disease in patients with good performance status, while chemo-/radiotherapy might be administered in patients with advanced disease or poor clinical conditions. Moreover, the extension of the disease might have influenced the strategy of care when planning a surgical approach compared to a non-surgical treatment, even if it is hard to estimate the real extent of this bias.

Moreover, only few information about the administration criteria of chemo-/radiotherapy were available in the included studies [23]. In the last years, several studies reported an increased number of patients (more than 80) [4,13,14,15,16], with better descriptions of patients’ characteristics and treatment indications aiming at reducing biases.

## 5. Conclusions

In conclusion, surgical management seems to be a valid and effective tool for recurrent thymoma patients and should always be considered as an option. This consideration is in agreement with the ESMO guidelines recommending a surgical approach whenever feasible [24] considering limited therapeutic potential offered by alternative therapeutic strategies (e.g., second and third chemotherapy lines). According to the results of our meta-analysis, surgical treatment is associated with a significantly better five and 10yrOS after recurrence, when matched against other care modalities. When exploring the prognostic impact of MG, histology, and the recurrence pattern (in particular in case of distant localization), data reported in the literature are not robust enough to draw any definitive conclusion. Therefore, further insight is needed.

When technically feasible and tolerable, surgery seem to be related to a favorable long-term outcome and should be considered as a fundamental part of a multimodal treatment in recurrent thymomas.

## Figures and Tables

**Figure 1 cancers-13-01559-f001:**
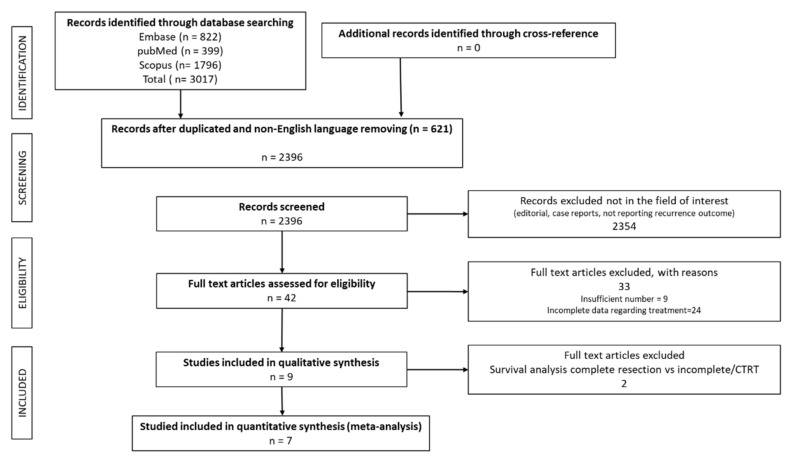
PRISMA diagram presenting the literature search and selection showing numbers of articles at each stage.

**Figure 2 cancers-13-01559-f002:**
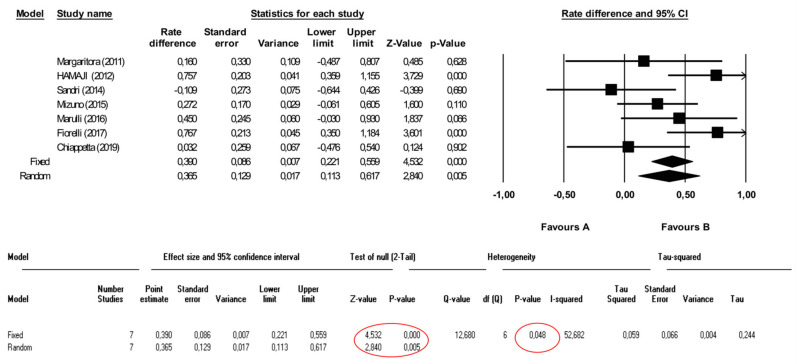
Forest plot showing the impact of surgery versus other treatments on overall survival.

**Table 1 cancers-13-01559-t001:** Studies’ characteristics.

Study	Number of Patients	Who Upstaging	Myasthenia Gravis	DFI (Months)	Surgery	Other Treatments	Complete Resection
RUFFINI (1997)	30	NR	22 (73.3%)	Mean 86 ± 45 (range 4–192)	16 (54%)	14 (46%)	10 (62.5%)
MARGARITORA (2011)	43	18 (60%)	40 (93%)	Mean 92.7 ± 77.8	30 (69.7%)	13 (30.3%) CT/RT 12 None 1	22 (73%)
HAMAJI (2012)	30	NR	13 (43%)	Median 61 (range 9–242)	20 (66.6%)	10 (33.4%)	18 (90%)
BAE (2012)	41	6 (40%)	22 (53.6%)	Median 52 (range 6–234	15 (36.6%)	26 (63.4%) CT/RT 25 Other 1	13 (87%)
SANDRI (2014)	81	25 (40.9%)	54 (66.7%)	Mean 86.5 ± 72.1	61 (61.3%)	20 (32.7%) CT/RT 14 Other 6	45 (72.5%)
MIZUNO (2015)	242	NR	54 (13.3%)	Mean 2.7 ± 2.3 (years)	119 (49.1%)	122 (50.9%)	Not reported for thymomas only
MARULLI (2016)	103	NR	63 (61.2%)	Median 50 (range 10–301)	73 (70.8%)	30 (29.2%) CT/RT 30	50 (68.5%)
FIORELLI (2017)	53	14 (37%)	30 (56%)	Mean 55 (range 38–69)	38 (71.7%)	15 (28.3%) CT/RT 13 Other 1	32 (60%)
CHIAPPETTA (2019)	155	24 (15.5%)	107 (69%)	Mean 78 ± 102	135 (87.1%)	20 (12.9%) CT/RT 19 Radio frequency 1	109 (70.4%)

WHO upstaging: change in histology from low to high grade, other treatments: DFI: disease free interval from primary thymectomy to recurrence appearance, CT: chemotherapy, RT: radiotherapy, other: ablation.

**Table 2 cancers-13-01559-t002:** Recurrence characteristics, surgical procedure and prognostic factors of the included studies * significant also at multivariable analysis. Patient numbers of Mizuno et al. are not reported due the concomitant presence of patients with thymic carcinoma.

Study	Recurrence Site (Number of Patients)	Surgical Procedure	Adjuvant Treatments (Number of Patients)	Favourable Prognostic Factors
RUFFINI (1997)	Loco-regional (26) Distant (4)	Resection Thoracotomy Sternotomy	Not reported	Local Recurrence Complete Resection
MARGARITORA (2011)	Pleura (25) Mediastinum (12) Lung (5), Liver And Bone (1)	Resection Thoracotomy Sternotomy	Not significant (*p* = 0.25)	Surgical Treatment Complete Resection
HAMAJI (2012)	Loco-Regional (28) Distant 2 (1 Liver, 1 Brain)	Resection	2 patients	Surgical Treatment Initial Masaoka Stage Complete Resection
BAE (2012)	Local (11) Regional (29) Distant (7)	Resection Thoracotomy Sternotomy	Radiotherapy (4) Chemoterapy (5) Chemoradiotherapy (2)	Complete Resection Histology (AB, B1)
SANDRI (2014)	Mediastinum (15) Pleura/Pericardium (47) Lung (13) Other Site (6)	Resection Thoracotomy Sternotomy	Pre or post operative chemo/radiotherapy: 15	Histology
MIZUNO (2015)	Pleura Lung Local Other	Resection	Not reported	Surgical Treatment Complete Resection
MARULLI (2016)	Local (17) Regional (63) Distant (14) Combined-Distant (9)	Resection Thoracotomy Sternotomy Laparotomy Single Pleural: Limited Pleural Resection Multiple Pleural Relapses: Partial Or Total Pleurectomy	Not significant (*p* = 0.87)	Complete Resection * Single Relapse * Initial Masaoka I-II * Loco-Regional Relapse * AB-B1-B2 Histology
FIORELLI (2017)	Local (13) Regional (26) Lung/Extrathoracic (11/3)	Resection VATS Thoracotomy	32 patients Chemotherapy alone 11 Radiotherapy alone 2 Chemotherapy/radiotherapy 19 Adjuvant therapy: 152 (115–161) months vs. 70 (28–149) months without (*p* = 0.03)	Ab-B1 Histology Complete Resection * Myasthenia Gravis Adjuvant Therapy
CHIAPPETTA (2019)	Local 21 Regional 111 Distant 23	Thoracotomy Sternotomy VATS Single Pleural: Limited Pleural Resection Multiple Pleural Relapses: Partial or Total Pleurectomy + HITOC	Adjuvant CT/RT 78 Not significant	Female Gender Myasthenia Gravis * Age Single Localization DFS > 36 Months *

**Table 3 cancers-13-01559-t003:** Survival outcome reported in different studies.

Author	OS		OS Surgery		OS Other Treatments		*p* Value	HR	95% CI	Note
	5Y	10Y	5Y	10Y	5Y	10Y				
RUFFINI (1997)	48	24	NR	NR	NR	NR	0.008	NR	NR	complete resection vs incomplete resection + other treatments
MARGARITORA (2011)	64	51	77	59	35	0	0.001	0.22	0.08—0.59	
HAMAJI (2012) *	50	NR	75.7	29.9	0	0	0.0002	NR	NR	Data reported in their successive meta-analysis
BAE (2012)	59.7	33.2	90.0	NR	40.7	NR	0.088	6.075	0.763–48.33	complete resection vs incomplete resection + other treatments
SANDRI (2014)	94.4	71.7	70.2	54.1	64.3	46.9	0.19	0.417	0.186–0.933	
MIZUNO (2015)	76.2	50	90.5	72.5	63.3	31.4	0.001	0.272	0.142– 0.521	
MARULLI (2016)	63	37	81	60	36	20	0.0001	7.65	3.07–19.10	
FIORELLI (2017)	52	32	92%	NR	15.3%	NR	0.0001	4.29	1.29–14.2	
CHIAPPETTA (2019)	70.2	44.4	70.5	49.1	67.3	15.1	0.064	1.91	0.96–3.79	

OS: Overall survival; Y: years; NR: Not reported. * some data are added in the meta-analysis published in 2014.

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
