# Peer review of "Which Is the Best Treatment in Recurrent Thymoma? A Systematic Review and Meta-Analysis"

_cancers, 2021, doi:10.3390/cancers13071559_

Round 1

Reviewer 1 Report

I think authors re-define surgical treatment for recurrent thymoma. I understand they are surgeons and would like to emphasize the importance of operation. But I think their description has to be neutral. From the text, I still don't understand how far thymoma is recurrent when the thymoma was operated. Figure text is still too small and not redable.

Author Response

We are grateful to Reviewer for his/her relevant comments and the work he/she has invested to help us improving our manuscript. We believe that his/her positive remarks and questions allowed us to significantly increase the interest of our manuscript.

Reviewer 1: I think authors re-define surgical treatment for recurrent thymoma. I understand they are surgeons and would like to emphasize the importance of operation. But I think their description has to be neutral. From the text, I still don't understand how far thymoma is recurrent when the thymoma was operated. Figure text is still too small and not redable.

Reply: First of all, thanks for your constructive suggestion. We have tried to reduce in the revised version of the manuscript the emphasis related to the surgical impact in terms of long-term outcome, as suugested.

Concerning the distinction between treatment of thymoma and treatment of oits recurrence, the disease-free interval (DFI, defined as the time between thymectomy and the occurrence of tumor relapse) was accurately defined in the text. In addition, since DFI were reported in the Table 1 for all studies considered in the meta-analysis, we include a better explanation of these data to further simplify the lecture by the Readers.

Regarding the figures, we tried to improve the resolution but the format of the article may reduce their quality. We believe that this new version could be better than before. Unfortunately, the format of “Cancer” does not allow to completely modify the figure sixe but forced us to fit it into a normal layout of the manuscript page. However, we hope that finally version will be more redable for the Readers.

Reviewer 2 Report

In this revised version, the authors have satisfactorily adressed the minor issues that had been raised by this reviewer.

However, there are still several typographic and grammatical errors (some of which had been pointed out in the previous review).

for example:

line 250 should read "primary" instead of "primitive" thymoma

line 317 should read "significantly"

line 359 should read "although it appears that..."

I leave it to the editor to decide whether this manuscript requires further language and spelling checks.

Author Response

We are grateful to Reviewer for his/her relevant comments and the work he/she has invested to help us improving our manuscript. We believe that his/her positive remarks and questions allowed us to significantly increase the interest of our manuscript.

Reviewer 2. In this revised version, the authors have satisfactorily addressed the minor issues that had been raised by this reviewer.

However, there are still several typographic and grammatical errors (some of which had been pointed out in the previous review).

for example:

line 250 should read "primary" instead of "primitive" thymoma

line 317 should read "significantly"

line 359 should read "although it appears that..."

I leave it to the editor to decide whether this manuscript requires further language and spelling checks.

Reply. Thanks for your constructive comments. We revised the language once again and revised the typographic and grammatical errors. We hope this new version of the manuscript could be acceptable for the Reviewer.

This manuscript is a resubmission of an earlier submission. The following is a list of the peer review reports and author responses from that submission.

Round 1

Reviewer 1 Report

1) Minor language:

a. Line 23: “thy” needs to be spelled out

b. Line 27: would use rarity (rather than numerosity)

c. Line 30: use individualizing instead of ‘individuate”

d. Line 57 use remains rather than ‘stays’

e. Line 81: reference PICOS or define

f.

2) The authors need to clarify “other treatments”

3) In line 95 the description of outcomes is not consistant with PICOS method. The outcome measure is survival and recurrence sites

4) Table 1. What is WHO upstaging mean? Define other treatments (chemotherapy, radiation, observation) How many had prior chemotherapy prior to surgery?

5) Table 1 and 2. I have reviewed the article by Hamaji and cannot reproduce the numbers used in this analysis.

6) Line 169. How are pleural recurrences considered: distant or intrathoracic? Pulmonary nodules are also "intrathoracic"

7) Line 200: not a "remarkable difference" when the p-value is 0.86

8) Line 210: Again make mention of the superiority over non-surgical treaments but don't specify numbers.

9) Line 233: not sure what to make of the surgical upgrading as it depends if there is central review. Non-specialists in thymic tumors as well as specialists have discordance.

10) Line 252: would be cautious and stating that this metanalysis "proves" the effectiveness of surgery. You allude to it later, but not emphasized is that there is a huge bias in selection factors. A patient with huge volume of tumor, serious co-morbid illnesses or otherwise poor surgical case will expect to have a worse outcome (regardless of surgery) compared to a PS=0 patient with low volume of disease.

11) Similarly in Line 310, the authors should be careful in overstepping conclusions outside of data presented.

Reviewer 2 Report

This paper deals with an interesting and relevant subject.

The methodology for the systematic review is well laid out (although I did not manage to find Appendix 1)

I would have some concerns about performing a formal meta-analysis on the data, because I am not convinced to what extent the studies are directly comparable. The information in the paper is not sufficient to allow sufficient jugdement as to this point.

Clearly the location and extent of tumour recurrence would be expected to have a major impact on treatment selection and treatment outcomes, and indeed, I suspect that the surgical and non-surgical populations here have inherently different disease. Little substantive information is provided to the reader in the results section (section 3,2 and 3,5,3 specifically) that allows any judgement to be made. It is necessary to elaborate and refine this section, which is key to the whole paper.

It would also be expected that many of the surgical patients in the included series had multimodality therapy. Although this would not directly affect the authors’ stated purpose of assessing the outcomes of patients who underwent surgery vs those who did not (regardless of any « combined » therapies) this nonetheless would have been of interest and practical relevance to the reader. It would also have been important to support some of the authors’ conclusions provided in the discussion (see below). It also would have been interesting to get some sense of how these patients were investigated (cross sectional imaging, PET scan, EBUS, etc.).

I understand that the authors may have been limited by the way these issues were initially adressed in the source publications. Nevertheless, I think it would have been possible, and indeed desirable, to discuss them in a much more substantive and informative way.

I feel that the authors’ affirmation that their study « proves the effectiveness » (Line 252) of surgical therapy requires some nuance.

The authors elaborate somewhat on recurrence patterns, the role of debulking, and HIPEC/adjuvant treatment in the discussion, but it reads more like an opinion than a formal analysis of the data. In addition, there seem to be some new data provided here that in my opinion should have appeared in the results section. As it stands, several of the authors’ conclusions, although they may or may not be valid, are not substantiated by their results. Among others, the authors mention the validity of operating distant or diffuse pleural disease (lines 258-60, 268-71); « completeness of the resection and the administration of integrated and multiple therapies plays a role » (lines 274-5); role of debulking (lines 281-83); role of adjuvant chemotherapy and multidisciplinary management (lines 291-303, 306-08).

Since the authors invested a lot of effort in their literature review, I think that there is an opporunity here to provide a structured, in depth analysis and substantive, thorough discussion of these different issues.

The role of MG in outcomes was adequately summarized. The characteristics of histology and their relation to outcomes was also adequately summarized.

There are some minor syntax issues throughout that should be revised for optimal clarity.

Additonal notes :

Line 145 : please mention Pubmed/Medline, Scopus, AND EMBASE (as this database was also included in the methods).

Figure 1 : « additonal records identified through other sources »; according to the methods section, sources were clearly defined as : Pubmed/Medline, Scopus, AND EMBASE; therefore, this box (« additonal records identified through other sources ») should be removed.

Table 1 :
include explanation of « WHO upstaging »; this is outlined in the text but should be included with the table.
« Recurrence complete resection » : I understand that this means the rate of recurrence in the surgical patients. It should be rephrased for clarity.

Author Response

please see attachement

Reviewer 3 Report

In this review, authors did meta-analysis and tried to define appropriate treatments for recurrent thymoma.

It is important to conclude surgical treatment is the best option.

But from their study, it is not clear whether with distant recurrence, still surgical treatment is the best option. The site of the recurrence may have an influence on treatment options. 

Figure 2, the texts are too small.

Reviewer 4 Report

In this meta-Analysis, Chiapetta et al. review the value of surgery in recurrent thymoma. This is a valuable source of information of a typical clinical problem in this specific field. The problems inherent to any meta-analysis dealing mostly with retrospective data are evident and were critically adressed and discussed by the authors.

Specific comments:

there are a number of grammatical errors that need critical editing (e.g. line 29 "indivituate", line 231 "primitive" instead of "primary" thymoma, line 334, etc.)

the limitations of retrospective studies and analysis of a limited number of papers become evident in the sections dealing with histology: "upgrading" (i.e. the morphological progression) of thymoma (paragraph 3.5.2) is a very rare event and the high percentages from the cited publications could also be explained by diagnostic or sampling errors. There are no accepted data to label type B2 thymomas as "low grade" (as opposed to type B3) (lines 232, 237)

 While it is of course at the discretion of the authors to cite individual opinions, the idea that only the cortical part of a thymoma may be responsible for a recurrence (line 337) is in stark contrast to all current expert concepts expressed e.g. in the WHO classification with very little evidence to support this hypothesis.

Round 2

Reviewer 2 Report

There remain several significant issues with the paper:

Briefly, in the results section the distinction between local and distant recurrence is very important. In section 3.2, we are given ranges gleaned from the reviewed studies, but not pooled results, which would have been more informative and more in keeping with the title (meta-analysis). I did not see the authors make any distinction or indeed any reference to the specific site of loco-regional recurrence in the reviewed studies, whether the original tumor bed or pleural metastases. Clearly, these would be expected to have significant implications with regard to treatment selection and outcomes.

Major issues remain with the discussion. Once again, the authors have presented new results, and have drawn conclusions without sufficient supportive data.
The following are examples :
The statement that « surgical treatment of thymoma seems to confer survival benefit even in the case of distant recurrence » is not substantiated by the data that the authors have presented.
Regarding the prognostic significance of the SPECIFIC SITE of distant metastatic relapse, no data are presented, except citing the Mizuno paper which included thymic carcinomas in their analysis.
« The pleura was the most common recurrence site, and surgery varied from resection of a single localization to extended pleurectomies associated with HITOC » : where is the data?
« Margaritora et al extensively described debulking surgery in 8 patients reporting a significantly inferior OS »; these are results that should have been included in the results section. There is no mention of this either in the text or the tables.
« an improvement in survival seems to be associated with adjuvant treatments (HITOC, chemo/rads to consolidate surgical results »; there is NO data presented to substantiate such a statement.
« Any CHANGE in histology did not influence the prognosis… »; where is the data?
« It appears that the histologic change shows a vector towards a more aggressive histology » : where is the data?

In conclusion, major issues with this paper remain. The strength of this paper is the number of patients, but the analysis fails to present the results in a meaningful way and the authors draw conclusions that are not substantiated by their data. Although the paper has potential, at this stage in the revision process it should be much further along. In my opinion, the manuscript does not meet the standard for publication.